# Antenatal corticosteroid administration and early school age child development: A regression discontinuity study in British Columbia, Canada

**Jennifer A. Hutcheon**[1]*, **Sam Harper**[2], **Jessica Liauw**[1], **M. Amanda Skoll**[1], **Myriam Srour**[3], **Erin C. Strumpf**[2,4]

1 Department of Obstetrics & Gynaecology, University of British Columbia, Vancouver, Canada,
2 Department of Epidemiology, Biostatistics, and Occupational Health, McGill University, Montreal, Canada,
3 Departments of Pediatrics and of Neurology and Neurosurgery, McGill University, Montreal, Canada,
4 Department of Economics, McGill University, Montreal, Canada

* jhutcheon@bcchr.ca

**Data Availability Statement:** Data cannot be shared publicly due to restrictions in the data sharing agreement under which the data were

## Abstract

### Background

There are growing concerns that antenatal corticosteroid administration may harm children's neurodevelopment. We investigated the safety of antenatal corticosteroid administration practices for children's overall developmental health (skills and behaviors) at early school age.

### Methods and findings

We linked population health and education databases from British Columbia (BC), Canada to identify a cohort of births admitted to hospital between 31 weeks, 0 days gestation (31+0 weeks), and 36+6 weeks, 2000 to 2013, with routine early school age child development testing. We used a regression discontinuity design to compare outcomes of infants admitted just before and just after the clinical threshold for corticosteroid administration of 34+0 weeks. We estimated the median difference in the overall Early Development Instrument (EDI) score and EDI subdomain scores, as well as risk differences (RDs) for special needs designation and developmental vulnerability (<10th percentile on 2 or more subdomains). The cohort included 5,562 births admitted between 31+0 and 36+6 weeks, with a median EDI score of 40/50. We found no evidence that antenatal corticosteroid administration practices were linked with altered child development at early school age: median EDI score difference of −0.5 [95% CI: −2.2 to 1.7] ($p = 0.65$), RD per 100 births for special needs designation −0.5 [−4.2 to 3.1] ($p = 0.96$) and for developmental vulnerability of 3.9 [95% CI: −2.2 to 10.0] ($p = 0.24$). A limitation of our study is that the regression discontinuity design estimates the effect of antenatal corticosteroid administration at the gestational age of the discontinuity, 34 + 0 weeks, so our results may become less generalisable as gestational age moves further away from this point.

obtained. The data underlying the results can be obtained through PopData BC (www.popdata.bc.ca/dataaccess), contact dataaccess@popdata.bc.ca". Statistical code used for our project can be accessed at: https://osf.io/2evfg/

**Funding:** This research was funded by the Canadian Institutes of Health Research (to JAH, ECS, and SH). The funders had no role in study design, data collection and analysis, decision to publish, or preparation of the manuscript.

**Competing interests:** The authors have declared that no competing interests exist.

**Abbreviations:** ALPS, Antenatal Late Preterm Steroids; ASTECS, Antenatal Steroids for Term Elective Caesarean Section; BC, British Columbia; CI, confidence interval; EDI, Early Development Instrument; ICD, International Classification of Diseases; RD, risk difference; RR, risk ratio; STROBE, Strengthening the Reporting of Observational Studies in Epidemiology.

## Conclusions

Our study did not find that that antenatal corticosteroid administration practices were associated with child development at early school age. Our findings may be useful for supporting clinical counseling about antenatal corticosteroids administration at late preterm gestation, when the balance of harms and benefits is less clear.

## Author summary

### Why was this study done?

- Antenatal corticosteroids are a medication given to pregnant women before a preterm delivery to help prevent breathing and other sorts of complications in their newborns.

- It has long been recommended that antenatal corticosteroids should be given to women at risk of preterm birth before 34 weeks of pregnancy, but new evidence suggests that newborns of women at risk of preterm birth at 34 to 36 weeks may also benefit from this medication.

- Because the breathing complications experienced by preterm births at 34 to 36 weeks are usually less serious in nature and less common, antenatal care providers are increasingly wanting to know if there are any long-term side effects of antenatal corticosteroids before administering the medication; however, there is limited information on the longer-term safety of antenatal corticosteroids for overall child developmental health.

### What did the researchers do and find?

- We linked population health and education databases from the province of British Columbia (BC), Canada to obtain early school age child development test scores in a cohort of children whose mothers were admitted to hospital for delivery between 31 and 36 weeks of pregnancy.

- During the time of our study, most women admitted for delivery before 34 weeks of pregnancy were given antenatal corticosteroids, while most women admitted after 34 weeks were not.

- Since pregnancies admitted for delivery just before and just after 34 weeks are similar in most ways except for in their chance of receiving antenatal corticosteroids, we compared the child development test scores of children whose mothers were admitted just before and just after 34 weeks of pregnancy to see if scores were lower in children who would have been exposed to antenatal corticosteroids according to clinical practice guidelines.

- We found no difference in child development test scores between children delivered just before and just after 34 weeks of pregnancy.

### What do these findings mean?

- Our findings suggest that a policy for routine administration of antenatal corticosteroids does not have a negative impact on overall child developmental health.

- Our findings can be used to support clinical counseling on the harms and benefits of antenatal corticosteroid administration at 34 to 36 weeks of pregnancy.

## Introduction

Clinical practice guidelines have long recommended that a single dose of antenatal corticosteroids should be administered to women with threatened preterm birth at 24 to 34 weeks gestation [1–3]. In 2016, the Antenatal Late Preterm Steroids (ALPS) trial examined the effectiveness of antenatal corticosteroids at late preterm gestation (34+0 to 36+6 weeks) [4]. They reported a 20% reduction in neonatal respiratory treatment or mortality in the intervention arm, providing evidence that corticosteroids are beneficial for preterm births of all gestational ages.

Yet, many jurisdictions have been hesitant to adopt routine administration of corticosteroids at late preterm ages [2,5]. Most cases of respiratory morbidity observed in the ALPS trial were transient tachypnea of the newborn, which is often considered a mild, self-limiting complication. Further, there are growing concerns that antenatal corticosteroids may have adverse effects on child neurodevelopment [6–9]. There is a plausible biological mechanism for harm through delayed myelination in the central nervous system [10,11], and corticosteroids could also alter neurodevelopment indirectly through their increased risk of hypoglycemia [4], which has been linked with impaired neurodevelopment in mid-childhood [12]. For some, the benefits of preventing a morbidity that is relatively benign and transient in nature may not outweigh the unknown longer-term risks of exposing fetuses to antenatal corticosteroids [13]. In the United Kingdom and Canada, updated clinical practice guidelines therefore only recommend that antenatal corticosteroids should be "considered" at late preterm ages, not "offered" [2,5].

Our overall goal was to generate evidence on the consequences of antenatal corticosteroid administration for longer-term child neurodevelopment. Because observational studies of medications such as antenatal corticosteroids are highly susceptible to confounding [13], we used a quasi-experimental design known as a regression discontinuity design [14], which exploits the fact that the decision to administer a medical intervention is often dictated by whether an individual is above or below an arbitrary clinical threshold along a biological continuum. With antenatal corticosteroids, Canadian clinical practice guidelines recommended (until 2018) that pregnancies with threatened preterm birth should be treated with corticosteroids up to 33+6 weeks, but not hours later, at 34+0 weeks [15]. Thus, infants immediately below the 34-week threshold are highly similar to those immediately above the threshold except for a higher probability of receiving antenatal corticosteroids, mimicking a randomized treatment assignment. In other words, we mimicked a pragmatic randomized trial in which 1 group had a policy to receive the intervention (antenatal corticosteroids), and the other group did not, but both groups are otherwise expected to be similar. We estimated the association between antenatal corticosteroid administration and longer-term child health by comparing the health outcomes of infants lying just above to those just below the threshold [14].

The objective of this study was to examine if antenatal corticosteroid administration practices were linked with children's overall development (skills and behaviors) at early school age in a large, population-based cohort, using a regression discontinuity design. We first examined

the extent to which the design could replicate the protective effect of antenatal corticosteroids on neonatal respiratory morbidity and mortality established in prior randomized trials [16].

## Methods

### Study population

Our study population was drawn from all live-born, singleton births in British Columbia (BC), Canada, with a delivery admission between 31+0 and 36+6 weeks, from April 1, 2000 to March 31, 2013. We identified our cohort using abstracted obstetrical and neonatal medical records contained in the BC Perinatal Data Registry, a validated population-based registry that contains records for >99% of births in the province [17]. Data quality are maintained through the use of provincially standardized medical record forms and standardized training of abstractors. Birth records were linked with BC Early Childhood Development early school age testing data (described below) and BC Vital Statistics records by Population Data BC, a multi-university organization that supports research using the province's administrative data [18–20]. Personal health numbers (a unique identifier for provision of universal health care), names, and birth dates were used for linkages. Children who moved out of province before age 4 (i.e., before being eligible to start school) were excluded. Analyses were conducted as specified in our prospective analysis plan (S1 Protocol). This study is reported as per the Strengthening the Reporting of Observational Studies in Epidemiology (STROBE) guideline (S1 Checklist).

### Ethics statement

The study was approved by the BC Children's Hospital Research Ethics Board (#H18-00620), which waived the requirement for individual-level informed consent.

### Gestational age estimation

Gestational age in days was calculated based on last menstrual period confirmed or revised by early ultrasound, using the algorithm used in clinical practice in Canada during the study period [21]. Births with no ultrasound confirmed- or revised-estimate of gestational age in days were excluded.

Day-specific estimates of gestational age based on ultrasound have been abstracted into the BC Perinatal Data Registry since April 1, 2008. Before this, ultrasound estimates of gestational age were abstracted in completed weeks only. For births prior to 2008, we calculated day-specific gestational age based on the date of the last menstrual period among women whose ultrasound-based estimate of gestational age (in weeks) was the same or within 1 week of their estimate based on last menstrual period (in weeks) if the dating scan was done <14 weeks, and within 2 weeks if done 14 to 20 weeks. We used gestational age at the time of maternal admission for the delivery admission as our unit of analysis (known as a "running variable" in the regression discontinuity design), which we hypothesized better approximated the timing of decision-making on antenatal corticosteroid administration than gestational age at delivery.

### Antenatal corticosteroids

During the study period, the recommended upper gestational age for antenatal corticosteroid administration in Canada was 33+6 weeks. Antenatal corticosteroid administration is available in the BC Perinatal Data Registry as a binary variable indicating the administration of corticosteroids during the delivery admission for the purpose of fetal lung maturation, but not administration during an antenatal hospitalization that did not result in delivery, therefore

underestimating overall administration. This known under-documentation did not affect our analyses, which used the gestational age discontinuity as the exposure, not corticosteroid status per se. That is, we relied instead on the assumption that antenatal corticosteroid administration recommendations were largely followed in practice, such that pregnancies admitted just before 34 weeks had a much higher change of being exposed to antenatal corticosteroids than those admitted just after (which we observed in our data).

## Early childhood development outcomes

BC schools conduct standardized assessments of early school age child development using the Early Development Instrument (EDI) during the first year of school (kindergarten), when children are aged 5 to 6 years [22]. The EDI provides a holistic assessment of children's development through 104 questions across 5 domains: physical health and well-being, social competence, emotional maturity, language and cognitive development, and communication skills. The EDI has been shown to be psychometrically sound and a good predictor of adult health, education, and social outcomes [23]. Each province-wide testing cycle takes 3 years to complete, meaning that in any given year, only approximately one-third of schools are tested. Students who started school in a year their school was not selected for testing (i.e., 2 of the 3 years within each cycle) do not have EDI scores. However, the EDI scores available (for approximately one-third of our birth cohort) provide a representative sample of children from across the province.

We examined the total EDI score (maximum = 50) and scores for each of the 5 EDI domains. We hypothesized that all domains were relevant to the detection of altered neurodevelopment. For example, in the physical health domain, questions such as "Can the child manipulate objects?" will capture concerns about fine motor skills and coordination, which are more subtle outcomes on the spectrum to cerebral palsy. Likewise, in the emotional maturity domain, questions such as "Is the child distractable, has trouble sticking to any activity?" assesses characteristics of attention deficit hyperactivity disorder. We examined a binary outcome of "developmentally vulnerable" (<10th population percentile on ≥2 domains), and a BC Ministry of Education designation of having special needs [22].

Infant or child deaths were assigned the lowest observed EDI score or occurrence of the adverse outcome. This was done to prevent selection bias introduced from differential losses to follow up (i.e., fewer cases of developmentally vulnerable children in the untreated group because infants who did not receive corticosteroids were less likely to survive infancy to develop the vulnerability) [24].

## Neonatal outcomes

We examined the extent to which the regression discontinuity design could replicate the protective effects of antenatal corticosteroids on neonatal respiratory morbidity/mortality previously demonstrated in prior randomized trials [16]. We created a composite outcome of either neonatal respiratory distress or all-cause death <28 days after birth, identified through International Classification of Diseases codes (ICD 10-CA: P22 or ICD 9-CA: 769, 770.6, 770.89) and vital statistics, respectively.

## Statistical analysis

The regression discontinuity design is based on an assumption that risks of adverse outcomes change smoothly over gestational age [14]. Because setting the threshold for antenatal corticosteroid recommendations at exactly 34 weeks is somewhat arbitrary, the design is based on the assumption that infants immediately below the 34-week threshold are highly similar to those

immediately above the threshold except for in the probability of receiving antenatal steroids. In the absence of any treatment effect, outcome rates should be smooth across the 34-week threshold; therefore, any abrupt change in the outcome immediately at 34 weeks can be attributed to antenatal corticosteroid administration practices (shown schematically in **S1 Fig**). In other words, the regression discontinuity design estimates the causal effect of the intervention at the point of the discontinuity (here, 34+0 weeks).

Standard regression discontinuity methods were used to estimate the size of any discontinuity in outcomes at 34+0 weeks [25]. The regression model included independent variables to control for underlying trends across gestational age (separately above and below the 34+0 week cut-off) and an indicator variable for being below the threshold at 34+0 weeks. This latter variable was our primary coefficient of interest, comparing differences in outcomes associated with being just above, versus just below, the 34+0 week threshold controlling for trends on either side. It is analogous to an intention-to-treat effect, because it reflects the impact of antenatal corticosteroid administration practices as they occur in a real-world setting (in which imperfect adherence with corticosteroid recommendations occurs because of incomplete courses, mistimed administration, and insufficient time to administer corticosteroids before delivery).

The model was implemented using quantile regression for continuous outcomes (estimating differences in the median EDI scores), and log-binomial models for binary outcomes. Observations were weighted according to their proximity to the 34+0 weeks threshold using a triangular kernel, such that observations closest to the discontinuity were assigned the greatest weight analytically and weights fell to near 0 at the limits of the data. Ninety-five percent confidence intervals (95% CIs) were generated using bootstrap resampling.

A number of sensitivity analyses were conducted to verify design assumptions [25]. We evaluated the gestational age specification using quadratic terms, with the best fit established using the Akaike Information Criterion and the Pseudo $R^2$. That is, all models assumed that risk of adverse neurodevelopmental outcomes changed smoothly across gestation (i.e., in the absence of an effect of corticosteroids at 34+0 weeks); the use of the quadratic term assessed whether this change occurred in a nonlinear manner. We also assessed whether there were any other factors changing discontinuously around the 34+0-week cut-off by plotting select maternal–fetal characteristics across gestational age. We examined different gestational age bandwidths (restricting to admissions between 32+0 to 35+6 weeks, and 33+0 to 34+6 weeks). We restricted analyses to children admitted for delivery after April 1, 2008. In children administered the EDI, we used multiple imputation to account for missing EDI values using multivariate imputation by chained equations with 10 imputed datasets. Models included study outcomes and maternal–fetal characteristics presented in Table 1. *p*-Values were approximated from bootstrapped 95% CI using the approach of Altman and Martin [26] and calculated using 2-sided, 2-sample tests of proportion and *t* tests to compare descriptive characteristics. A *p*-value less than 0.05 was considered statistically significant.

## Power estimation

We performed simulation studies to ensure that we had sufficient statistical power. Using the fixed sample size of eligible preterm births in the BC Perinatal Data Registry (and assuming EDI scores were only available for one-third of infants due to 3-year testing waves), the gestational age distribution of births within this age window in BC, the mean and standard deviation of EDI scores among preterm births [27], and the underlying association between gestational age and child development scores [28], we created multiple simulated datasets and ran regression discontinuity analyses to assess our power to detect different treatment effects

**Table 1. Descriptive characteristics of the neonatal cohort and those with EDI early school age testing in BC, Canada, 2000 to 2013.**

| Maternal–fetal characteristic | Neonatal cohort (*n* = 15,741) mean ± standard deviation or *n* (%) | EDI cohort (*n* = 5,562) mean ± standard deviation or *n* (%) |
|---|---|---|
| Maternal age (years) | 31 ± 6 | 31 ± 6 |
| Nulliparity | 7,767 (49) | 2,753 (50) |
| Smoking in pregnancy | 1,821 (12) | 714 (13) |
| Pre-pregnancy BMI (kg/m²)* | 25 ± 5 | 25 ± 5 |
| Hypertensive disorder of pregnancy | 2,445 (16) | 869 (16) |
| Diabetes in pregnancy | 2,708 (17) | 917 (16) |
| Cesarean delivery | 5,968 (38) | 2,078 (37) |
| Labor induction | 4,055 (26) | 1,419 (26) |
| Male fetus | 8,882 (56) | 3,163 (57) |
| Birthweight (grams) | 2,672 ± 534 | 2,669 ± 536 |
| Gestational week at admission for delivery | | |
| 31 | 528 (3) | 198 (4) |
| 32 | 794 (5) | 303 (5) |
| 33 | 1,164 (7) | 405 (7) |
| 34 | 1,937 (12) | 663 (12) |
| 35 | 3,546 (23) | 1,291 (23) |
| 36 | 7,772 (49) | 2,702 (49) |
| 5-minute Apgar score <7 | 1,216 (8) | 498 (9) |
| Neonatal respiratory morbidity or mortality | 3,257 (21) | 1,157 (21) |

*Among 11,533 women in full cohort and 3,968 women in EDI cohort with available BMI.

BC, British Columbia; BMI, body mass index; EDI, Early Development Instrument.

(i.e., magnitude of the difference in EDI scores before and after the 34+0 week discontinuity). From this, we estimated that we would have >90% power to detect an effect size of 0.2 standard deviations or greater, suggesting we would be able to identify even small effects of the medication.

## Results

### Study population

There were 526,525 singleton live births between April 1, 2000 and March 31, 2013. We excluded children who left the province before school-entry age and pregnancies with no day-specific estimate of gestational age confirmed or revised with early ultrasound (see **S2 Fig** for flow of participants). **S1 Table** compares the characteristics of pregnancies included versus excluded due to lack of a day-specific, ultrasound-based estimate of gestational age. As expected, the primary difference between included and excluded pregnancies was calendar time: 71% of admissions post-April 1, 2008 were included, versus only 46% of admissions prior to this date. Excluded pregnancies were also modestly younger (1 year), more likely to be parous (44 versus 47%) and more likely to smoke (13 versus 8%) than those included in our cohort, but there were no clinically important differences in pre-pregnancy body mass index, maternal comorbidities, birthweight, preterm birth, or neonatal outcomes. All differences in descriptive characteristics were statistically significant (*p* < 0.0001), likely reflecting our large sample size. The difference in neonatal respiratory morbidity or mortality was not statistically significant (*p* = 0.50).

Among eligible births with a day-specific, ultrasound-confirmed estimate of gestational age, 15,741 were admitted for the delivery admission between 31+0 and 36+6 weeks. Of these,

5,562 (35%) had EDI scores available (due to 3-year testing waves described above) or were a child death (46 neonatal deaths and 96 postneonatal deaths). As shown in **Table 1**, characteristics of infants for whom EDI scores were available were not meaningfully different than those in the total neonatal cohort. Differences between included and excluded births were not statistically significant (all $p > 0.05$), with the exception of maternal age ($p = 0.001$) and smoking in pregnancy ($p = 0.002$).

## Administration of antenatal corticosteroids

**Fig 1** shows the pattern of antenatal corticosteroid administration by gestational age. As expected, the proportion of infants administered antenatal corticosteroids was substantially higher (45 percentage points, 47% versus 2%) before versus after 34+0 weeks (238 days). The decrease occurred primarily over a 4-day period between 33+4 and 34+0 weeks.

## Neonatal respiratory morbidity and mortality

The risk of neonatal respiratory morbidity/mortality by gestational age is shown in **S3 Fig**. As expected, risks decreased with advancing gestation, from over 50% at 31+0 weeks to less than 10% at 36+6 weeks. However, risks after 34+0 weeks were shifted to systematically higher values than those expected based on the rate of decrease before 34+0 weeks. The age at which risk began to plateau, at approximately 33+4 weeks, coincided with the age at which corticosteroid administration began to decrease (**Fig 1**). Our regression discontinuity model estimated that the higher rates of antenatal corticosteroid administration before 34+0 weeks were associated

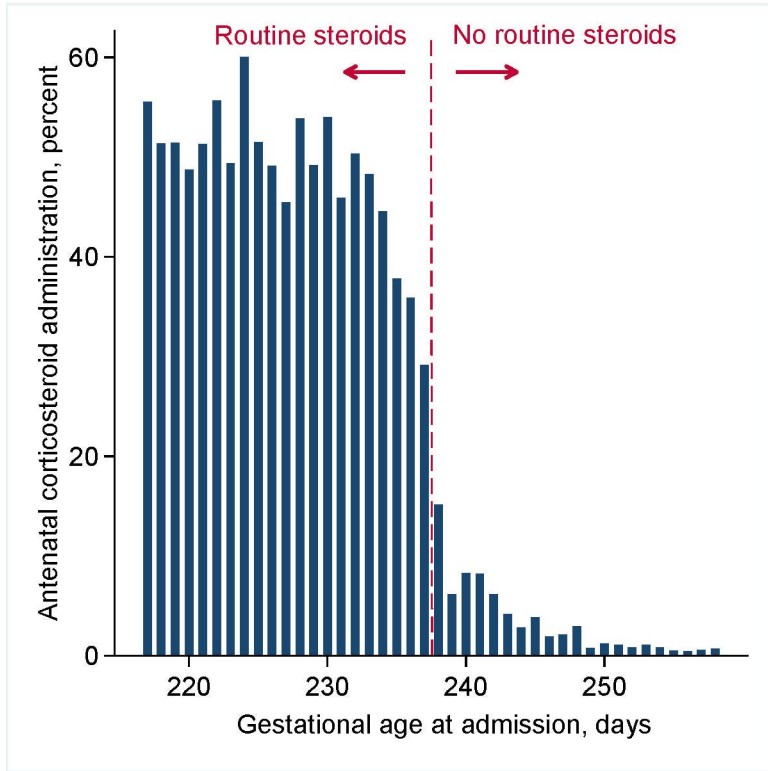

**Fig 1. Antenatal corticosteroid administration among 15,741 births in BC, Canada, 2000 to 2013.** Vertical dashed line indicates the upper limit of recommended administration, 33+6 weeks (237 days). BC, British Columbia.

with a risk ratio for neonatal respiratory morbidity/mortality of 0.81 [95% CI: 0.71 to 0.92] ($p$ = 0.001), or 5.4 fewer cases per 100 deliveries [95% CI: 2.3 to 8.8] ($p$ = 0.001).

## Child development testing

Fig 2 shows that antenatal corticosteroid administration practices were not associated with total EDI scores (median difference of −0.5 [95% CI: −2.2 to 1.7] ($p$ = 0.65) at the discontinuity of 34+0 weeks; Table 2). Analyses for each of the 5 domains revealed similar results and showed no clinically or statistically meaningful differences with corticosteroid administration practices (all differences in median scores <0.5) (S4 Fig). The point estimate for developmental vulnerability was suggested a small increase in risk (3.9 excess cases per 100 deliveries [95% CI: −2.2 to 10.0]; risk ratio (RR) 1.16 [95% CI: 0.91 to 1.45]), but the 95% CI provided evidence consistent with a null effect (Fig 3; $p$ = 0.24). There was no effect of antenatal corticosteroid administration practices and special needs designation (excess risk per 100 deliveries −0.5 [95% CI: 4.2 to 3.1], RR 0.9 [95% CI: 0.6 to 1.4]; $p$ = 0.96; Fig 4).

## Sensitivity analyses

None of the sensitivity analyses were inconsistent with our primary findings. There was no evidence of violations in the regression discontinuity model assumption that no other variables change abruptly at the point of the discontinuity (S5 Fig). Using multiple imputation to account for missing EDI results produced estimates for total EDI score, developmental

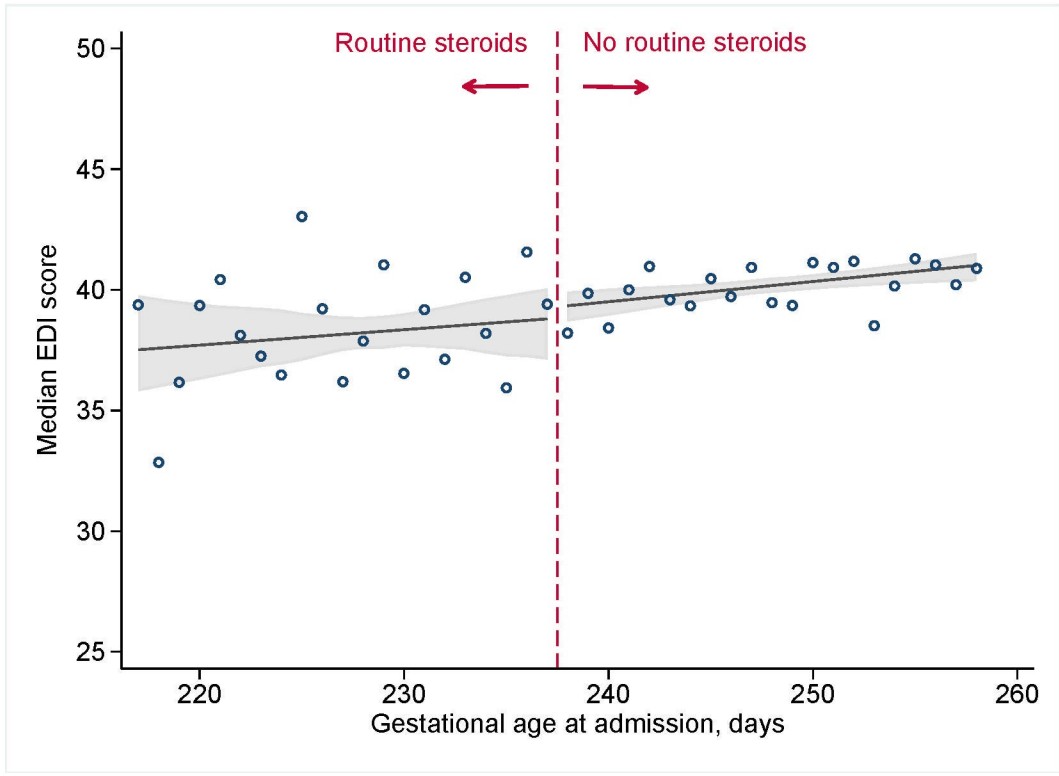

**Fig 2. EDI scores among 5,562 kindergarten-aged children in BC, Canada admitted for the delivery admission between 31+0 and 36+6 weeks of gestation, 2000 to 2013.** Vertical dashed line indicates the upper limited of recommended administration, 33+6 weeks (237 days). Circles indicate observed day-specific values, and solid line indicates the smoothed estimate with 95% CI (shaded area). BC, British Columbia; CI, confidence interval; EDI, Early Development Instrument.

**Table 2. Estimated "intention-to-treat" effect of antenatal corticosteroid administration practices among 5,562 children in BC, Canada admitted for the delivery admission between 31+0 and 36+6 weeks of gestation, 2000 to 2013.**

| Outcome[1] | Median [interquartile range] or *n* (%)* | Estimated effect of corticosteroid administration practice (<34 weeks of gestation vs. reference of ≥34 weeks) | | | |
|---|---|---|---|---|---|
| | | Absolute difference in median scores [95% CI] | *p*-value | | |
| Total EDI score (/50) | 40.1 [32.7, 45.3] | −0.5 [−2.2, 1.7] | 0.65 | - | |
| Communication skills score (/10) | 6.9 [5, 10] | −0.4 [−1.3, 0.8] | 0.51 | - | |
| Emotional maturity score (/10) | 7.8 [6.3, 9.8] | −0.2 [−0.6, 0.2] | 0.43 | - | |
| Language and cognitive development score (/10) | 8.8 [6.9, 9.6] | 0.5 [−0.3, 0.6] | 0.52 | - | |
| Physical health and well-being score (/10) | 8.1 [5.4, 9.6] | 0.0 [−0.2, 0.1] | 0.68 | - | |
| Social competence score (/10) | 8.5 [6.4, 9.6] | −0.2 [−0.7, 0.4] | 0.56 | - | |
| | | Excess cases per 100 births [95% CI] | | RR [95% CI] | *p*-value |
| Developmentally vulnerable | 1,309 (23.7) | 3.9 [−2.2, 10.0] | 0.24 | 1.2 [0.9, 1.5] | 0.21 |
| Special needs designation | 417 (7.5) | −0.5 [−4.2, 3.1] | 0.96 | 0.9 [0.6, 1.4] | 0.80 |

[1]Total EDI score missing in 50 children (0.9%), Communication skills score in 32 children (0.6%), Emotional maturity score in 68 children (1.2%), Language and cognitive development score in 61 children (1.1%), Physical health and well-being score in 39 children (0.7%), Social competence score in 51 children (0.9%), Developmental vulnerability in 29 children (0.5%), and Special Needs status in 16 children (0.3%).

BC, British Columbia; CI, confidence interval; EDI, Early Development Instrument; RR, risk ratio.

vulnerability, and special needs designation were nearly identical to our primary analyses (−0.5 [95% CI: −2.1 to 1.6] (*p* = 0.60), 1.16 [95% CI: 0.9 to 1.4] (*p* = 0.25), and 1.0 [95% CI: 0.6 to 1.4] (*p* = 0.77), respectively). Multiply imputed models for each of the 5 domains failed to converge in some of the bootstrapped iterations, precluding the calculation of 95% CIs; however, all point estimates were within 0.2 of the primary estimates. Changing the gestational age window of births included in the analyses, or restricting to admissions post-April 1, 2008, did not meaningfully impact our conclusions (**S2 and S3 Tables**).

## Discussion

### Statement of principal findings

In this large, population-based cohort, we found no evidence that antenatal corticosteroids administration practices were associated with child development scores at early school age. Our use of the regression discontinuity design was supported by its ability to replicate the protective effect of antenatal corticosteroids on neonatal respiratory morbidity and mortality previously established in meta-analysis of randomized trials (reductions in respiratory distress syndrome of 34% [95% CI: 23 to 44] in all trials [16] and 26% [95% CI: 9 to 39] at late preterm [29] versus 19% [95% CI: 9 to 28] in our study). These findings may provide reassurance to clinicians and patients concerned about adverse longer-term consequences of corticosteroid administration.

### Comparison with other studies

Numerous studies have investigated the longer-term neurodevelopmental safety of antenatal corticosteroids, but conclusions have been limited by methodological challenges. A 31-year follow-up study of the 1970s Auckland steroid trial found no differences in cognitive functioning or working memory between corticosteroid and placebo groups [30]. However, the study included only 192 of 988 neonatal survivors (19%), creating the potential for significant

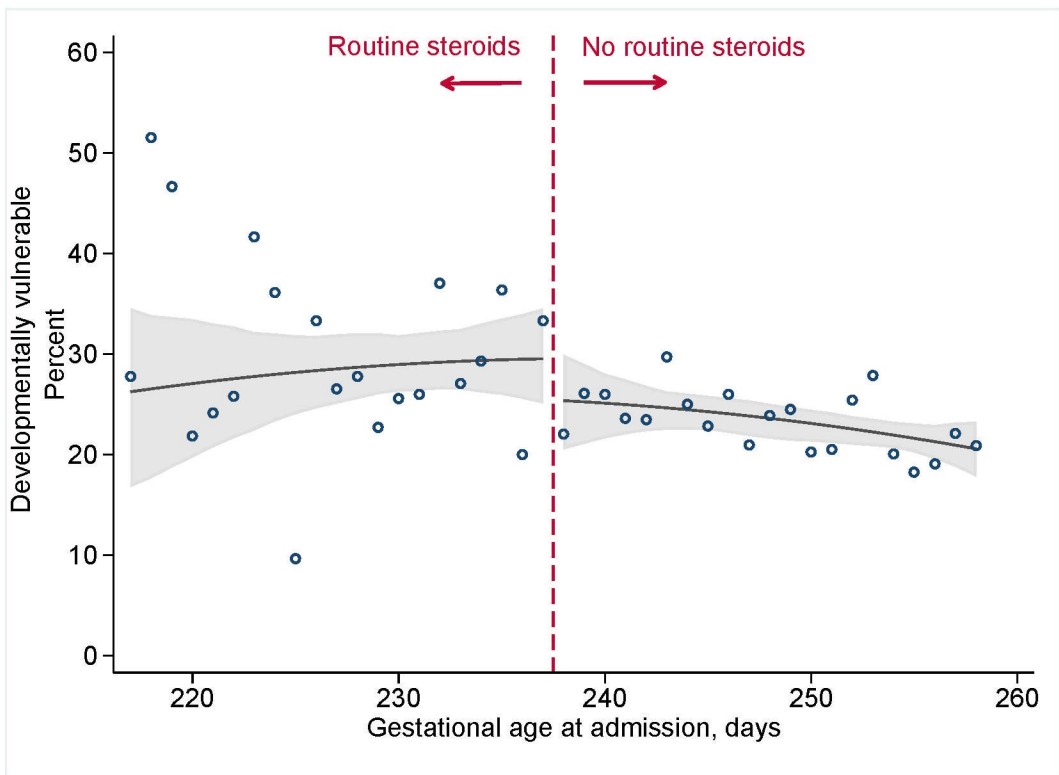

**Fig 3. Developmental vulnerability among 5,562 kindergarten-aged children in BC, Canada, admitted for the delivery admission between 31+0 and 36+6 weeks of gestation, 2000 to 2013.** Vertical dashed line indicates the upper limited of recommended administration, 33+6 weeks (237 days). Circles indicate observed day-specific values, and solid line indicates the smoothed estimate of risk with 95% CI (shaded area). BC, British Columbia; CI, confidence interval.

selection bias. A 20-year follow-up study of a subsequent Dutch trial found no differences in cognitive functioning or any other measures of health according to corticosteroid assignment status [31]. However, only 81 individuals were included, making it likely underpowered to detect even moderate-to-large effects. There were likewise no differences between groups in a follow-up of the 1976 United States National Heart, Lung, and Blood Institute randomized trial (27.0% of children in the treatment group had an abnormal or suspect McCarthy General Cognitive Index versus 28.4% in the placebo group) [32]. However, follow-up did not persist past the first 36 months of life, before more subtle neurodevelopmental effects may manifest, and sample size was still relatively modest ($n = 406$).

A follow-up of the Antenatal Steroids for Term Elective Caesarean Section (ASTECS) trial evaluating corticosteroid administration prior to planned cesarean delivery at term found no significant differences between groups in the school-based strengths and difficulties question-naire [33]. It did, however, find that children in the betamethasone arm were more likely to be ranked in the lower quarter of academic activity by their school than those in the control arm (17.7% versus 8.5%, respectively). However, follow-up was only 51% for questionnaire-based outcomes and 44% for school assessment data, with differential follow-up by both treatment assignment and study outcomes.

A 2015 meta-analysis found lower risks of neurodevelopmental outcomes such as cerebral palsy and low psychomotor development index score among those who received corticoste-roids compared with those who did not [34]. However, the majority of studies in the meta-analysis were observational studies that used conventional regression adjustment to control

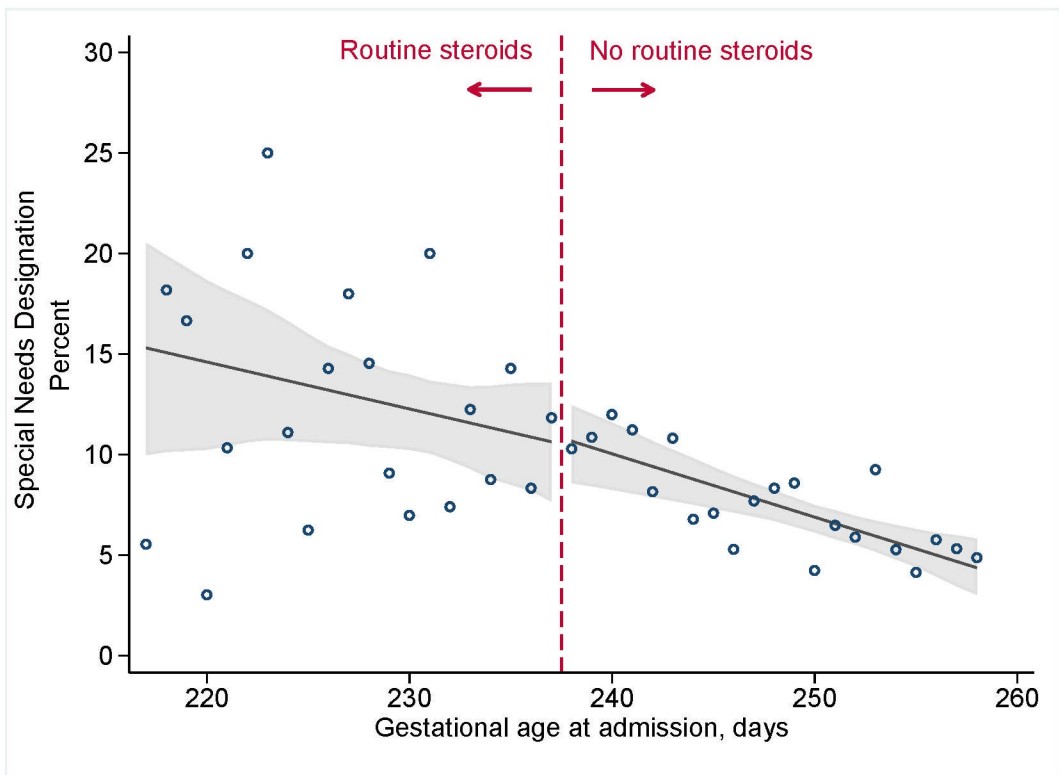

**Fig 4. Ministry of Education designation as special needs among 5,562 kindergarten-aged children in BC, Canada, admitted for the delivery admission between 31+0 and 36+6 weeks of gestation, 2000 to 2013.** Vertical dashed line indicates the upper limited of recommended administration, 33+6 weeks (237 days). Circles indicate observed day-specific values, solid line indicates the smoothed estimate of risk with 95% CI (shaded area). BC, British Columbia; CI, confidence interval.

for confounding. As a result, considerable potential for confounding exists because of the difficulties in fully accounting for differences in risk profiles of pregnancies that do versus do not receive antenatal steroids.

Our findings are consistent with this body of evidence largely suggesting that corticosteroids do not cause neurodevelopmental harm, but considerably strengthen the level of evidence behind this conclusion, through our use of a quasi-experimental design that minimized confounding in a large, population-based sample.

### Strengths and weaknesses

The primary strength of this study is that it provides robust real-world evidence on the safety of antenatal corticosteroid administration practices. Although randomized trial evidence is often viewed as the gold standard for medical decision-making, findings may reflect treatment effects in small study populations that are often highly selected, with optimal conditions for treatment administration and limited follow-up [35]. Evidence from our study complements randomized trial-based findings by estimating effects of corticosteroid administration practices in a broad population mix that reflects administration regimes given in day-to-day care. Population linkages helped to maximize follow-up rates over longer periods of time, which is a common challenge with investigator-led data collection, and provided a sample size twice as large as the largest corticosteroids trial to date.

A relatively large fraction of the cohort was excluded due to lack of a day-specific, ultrasound-confirmed estimate of gestational age. However, we believe that a majority of these exclusions were due to limitations of the Data Registry variable definitions rather than systematic differences between women who did versus did not have a day-specific ultrasound estimate. Births before April 1, 2008 were much more likely to be excluded, as the Data Registry only collected week-, rather than day-specific ultrasound estimates before this date. Further, the variable definition for gestational age by ultrasound specifies that only dating scans done prior to 20 weeks should be abstracted. A previous validation study [17] found that of the pregnancies with missing information on the dating ultrasound, 53.5% had a dating ultrasound at 20 to 23+6 weeks (the recommended upper limit for ultrasound gestational age dating). That is, half of the women with missing ultrasound-based estimates of gestational age were dated by ultrasound in practice, the information simply was not abstracted into the Data Registry.

This validation study [17] further found that of those with a missing ultrasound-based estimate of gestational age, 10.6% were also missing antenatal forms in their charts. We speculate that the lack of ultrasound information for these women likely reflects issues in the transfer of records to the delivery hospital, rather than a lack of ultrasound estimation. Thus, although excluding births without a day-specific, ultrasound-confirmed estimate of gestational age reduced our sample size, we do not believe this was likely to have introduced selection bias because pregnancies excluded on the basis of calendar year (i.e., before or after 2008) or on the basis of a failure to transfer hospital records would be unlikely to have systematic difference regarding the effect of antenatal corticosteroids on child development. The primary consequence of a reduced sample size would be imprecise estimates (i.e., wide 95% CIs), but our primary findings had good precision. Given the importance of accurate gestational age information for the regression discontinuity design, we opted to exclude these cases rather than retain them through multiple imputation or use of last menstrual period dating, which could introduce measurement error through misestimated gestational age.

Data on the dose or timing of antenatal corticosteroid administration were unavailable. However, date of maternal admission to hospital is a reasonable proxy for gestational age of administration as the need for corticosteroids is typically established shortly after admission, and our data show a clear discontinuity at 34 weeks (S3 Fig). Our assumption that decisions on antenatal corticosteroid administration are made around the time of maternal admission is informed by the clinical experience of coauthors MAS and JL, but cannot be empirically tested in our data. Further, in the ALPS trial [4], 3,269 of 24,333 (13.5%) pregnancies screened for eligibility at 34 to 36 weeks had received antenatal corticosteroids earlier in pregnancy. If a sizeable fraction of infants in our cohort admitted for delivery at 34 to 36 weeks had also received steroids earlier in pregnancy, this could introduce bias to our findings (since our analysis assumed that infants admitted after 34 weeks had a low likelihood of exposure to antenatal corticosteroids). It is reassuring, however, that our estimates of effect for neonatal respiratory morbidity/mortality were compatible with those reported from randomized trials [29], suggesting the magnitude of any such bias is likely small. In some situations, the regression discontinuity design can produce an estimate that accounts for nonadherence to practice guidelines (i.e., the effect of corticosteroids under optimal administration). However, this approach requires accurate individual-level records of antenatal corticosteroid administration, which we lacked. We therefore limited our analyses to the overall "intention-to-treat" effect of antenatal corticosteroid administration practices.

Finally, the regression discontinuity design only allows inference on the effects of antenatal steroid administration at the point of the discontinuity: 34+0 weeks [14]. It is possible that the association between antenatal corticosteroids and child neurodevelopment differs according to gestational age (especially at term) [6,9], and the generalizability of our findings may

decrease as gestational age moves farther from 34+0 weeks. Given that corticosteroid administration at 34 to 36 weeks is controversial [7,9,13] and high-quality evidence on longer term effects from any gestational age is currently lacking, we nevertheless believe that our findings are still an important contribution. Further, a small fraction of women presenting at preterm ages do not deliver preterm (in the ALPS trial, 16% of women presenting to hospital at 34 to 36 weeks delivered after 36+6 weeks [4]). Our conclusions should not be applied to such pregnancies, and neurodevelopmental effects may differ in this group.

## Conclusions

The results of this study suggest that antenatal corticosteroid practices are not linked with increased risks of adverse child development at early school age. These estimates may be useful to inform evidence-based discussions of harms and benefits for care providers and families contemplating administration of antenatal corticosteroids at late preterm ages.

## Supporting information

**S1 Fig. Flow of participants.**
(DOCX)

**S2 Fig. Schematic of the regression discontinuity design assuming antenatal corticosteroids increases the risk of adverse developmental outcome (hypothetical data).** The vertical dashed line indicates the upper gestational age cut-off for routine administration of antenatal corticosteroids (33+6 weeks of gestation). The terms "β1" and "β2" estimate the underlying trend in risk of adverse neurodevelopmental outcomes associated with gestational age (as risks of most adverse outcomes decrease with advancing gestational age) before and after the cut-off, respectively. The term "β3" is the primary estimate of interest, which estimates if there is a "jump" or level change in risk at the gestational age at which corticosteroids stop being administered routinely (i.e., at the point of discontinuity in corticosteroid treatment practices). If antenatal corticosteroid administration causes adverse neurodevelopmental outcomes, we would expect a decrease in risk at 34+0 weeks, reflecting that risks dropped once fetuses were no longer exposed to antenatal corticosteroids.
(DOCX)

**S3 Fig. Neonatal respiratory morbidity or mortality among 15,741 births in British Columbia, Canada, 2000 to 2013.** Vertical dashed line indicates the upper limited of recommended administration, 33+6 weeks (237 days). Circles indicate observed day-specific risks of neonatal respiratory morbidity/mortality; solid circles highlight the gestational ages coinciding with reduced antenatal corticosteroid administration. Solid line indicates the smoothed estimate of risk with 95% confidence interval (shaded area).
(DOCX)

**S4 Fig.** Early Development Instrument (EDI) test results among 5,562 kindergarten-aged children in British Columbia, Canada, admitted for the delivery admission between 31+0 and 36+6 weeks of gestation, 2000 to 2013, for EDI subdomains of (a) communication skills, (b) emotional maturity, (c) language and cognitive development, (d) physical health and well-being, and (e) social competence. Vertical dashed line indicates the upper limited of recommended administration, 33+6 weeks (237 days).
(DOCX)

**S5 Fig. Distribution of select maternal–fetal characteristics across gestational age to verify the assumption that variables are continuous across the 34+0 weeks discontinuity in**

**antenatal corticosteroid recommendations.**
(DOCX)

**S1 Table. Characteristics of pregnancies excluded due to lack of day-specific, ultrasound-based gestational age data among singleton births in British Columbia, Canada, April 1, 2000 to March 31, 2013.**
(DOCX)

**S2 Table. Effect of changing the gestational age window around the 34+0 week discontinuity when estimating the effect of antenatal corticosteroid administration practices among 5,562 children in British Columbia, Canada, 2000 to 2013.**
(DOCX)

**S3 Table. Effect of restricting to births admitted after April 1, 2008 when estimating the effect of antenatal corticosteroid administration practices among 5,562 children in British Columbia, Canada.**
(DOCX)

**S4 Table. Characteristics of children with versus without missing total Early Development Instrument (EDI) scores in British Columbia, Canada, 2000 to 2013.**
(DOCX)

**S1 Protocol. Preanalysis plan (January 27, 2020).**
(PDF)

**S1 Checklist. STROBE checklist for observational cohort studies.**
(PDF)

## Acknowledgments

We thank Tim Choi from Population Data BC, Vancouver, Canada for his invaluable assistance in data access.

All inferences, opinions, and conclusions drawn in this study are those of the authors and do not reflect the opinions or policies of the Data Stewards.

## Author Contributions

**Conceptualization:** Jennifer A. Hutcheon, Sam Harper, Jessica Liauw, M. Amanda Skoll, Myriam Srour, Erin C. Strumpf.

**Formal analysis:** Jennifer A. Hutcheon, Sam Harper, Jessica Liauw, M. Amanda Skoll, Myriam Srour, Erin C. Strumpf.

**Funding acquisition:** Jennifer A. Hutcheon, Sam Harper, M. Amanda Skoll, Myriam Srour, Erin C. Strumpf.

**Methodology:** Jennifer A. Hutcheon, Sam Harper, Erin C. Strumpf.

**Project administration:** Jennifer A. Hutcheon.

**Writing – original draft:** Jennifer A. Hutcheon.

**Writing – review & editing:** Sam Harper, Jessica Liauw, M. Amanda Skoll, Myriam Srour, Erin C. Strumpf.

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
