## [Editor Report · Decision Letter 0]

18 Jun 2020

Dear Dr Hutcheon, 

Thank you for submitting your manuscript entitled "Antenatal corticosteroid administration and kindergarten-age child development: a regression discontinuity study" for consideration by PLOS Medicine.

Your manuscript has now been evaluated by the PLOS Medicine editorial staff, as well as by an academic editor with relevant expertise, and I am writing to let you know that we would like to send your submission out for external peer review.

Kind regards,

Caitlin Moyer, Ph.D.,

Associate Editor

PLOS Medicine

---

## [Decision Letter · Decision Letter 1]

19 Aug 2020

Dear Dr. Hutcheon,

Thank you very much for submitting your manuscript "Antenatal corticosteroid administration and kindergarten-age child development: a regression discontinuity study" (PMEDICINE-D-20-02762R1) for consideration at PLOS Medicine. 

Your paper was evaluated by a senior editor and discussed among all the editors here. It was also discussed with an academic editor with relevant expertise, and sent to three independent reviewers, including a statistical reviewer. The reviews are appended at the bottom of this email and any accompanying reviewer attachments can be seen via the link below:

[LINK]

In light of these reviews, I am afraid that we will not be able to accept the manuscript for publication in the journal in its current form, but we would like to consider a revised version that addresses the reviewers' and editors' comments. Obviously we cannot make any decision about publication until we have seen the revised manuscript and your response, and we plan to seek re-review by one or more of the reviewers. 

We expect to receive your revised manuscript by Sep 09 2020 11:59PM. Please email us (plosmedicine@plos.org) if you have any questions or concerns.

We look forward to receiving your revised manuscript. 

Sincerely,

Caitlin Moyer, Ph.D.

Associate Editor 

PLOS Medicine

plosmedicine.org

1.Please include the study population in the title, such as: “Antenatal corticosteroid administration and kindergarten-age child development: a regression discontinuity study in British Columbia, Canada” or similar.

2.Data availability: Thank you for noting that “The data underlying the results can be obtained through

PopData BC (www.popdata.bc.ca/)” If possible, please provide a more direct link or contact address for the specific data set.

3. Please structure your abstract using the PLOS Medicine headings (Background, Methods and Findings, Conclusions).

4. Abstract: In the last sentence of the Objective/Background section, can you please be more specific with the goal of the study, in terms of outcome? “We investigated the safety of antenatal corticosteroid administration practices for child development at kindergarten age.”

5.Abstract: Line 41 (as well as in the Title and throughout text): When referring to “kindergarten age” can you please define or replace with a more universal term or measure of the childrens’ ages? (e.g. 5 years of age).

6. Abstract: Line 44-45 (and in the Introduction at Lines 83-84: Please remove the descriptor "a quasi-experimental design known as" because we feel that most readers will recognize the regression-discontinuity design.

7. Abstract: Methods and Findings: Please include both the 95% CIs and the p values associated with all reported findings. 

8. Abstract: Methods and Findings: As the final sentence in this section, please state the main limitation(s) of the study.

9. Abstract: Conclusions: Please revise the first sentence to: “Our study suggests that antenatal corticosteroid administration practices do not appear to affect child development at kindergarten age.” 

10. Author Summary: At this stage, we ask that you include a short, non-technical Author Summary of your research to make findings accessible to a wide audience that includes both scientists and non-scientists. The Author Summary should immediately follow the Abstract in your revised manuscript. This text is subject to editorial change and should be distinct from the scientific abstract. Please see our author guidelines for more information: https://journals.plos.org/plosmedicine/s/revising-your-manuscript#loc-author-summary

11. In text citations: Please place the in-text citation within square brackets before the punctuation mark, like this: [1].

12. Introduction: Line 63: Please define the acronym ALPS at first use.

13. Introduction: Line 95: Please revise the sentence: “ The objective of this study was to examine the safety of antenatal corticosteroid administration” to change “safety” to more clearly describe the study outcomes under investigation.

14. Methods: Please specify the manner of informed consent, or if the requirement for consent was waived, and if so by what institution.

15. Methods: Line 148-149: Can you please clarify whether all the children were tested in “kindergarten” or at age 5, or at what ages or range the children were tested? “The province is tested in 3-year waves, so only approximately one third of children are tested in a given year.”

16. Methods: Line 189: Please replace the term “compliance” with “adherence” where it is used to refer to treatment adherence.

17. Results: Please provide numbers and percents for missing data within the results or tables.

18. Results: Line 211: and Table S1: Were there statistically significant differences between included/excluded children/pregnancies?

19. Results: 221-222: Please clarify “meaningful differences” if “statistically significant differences” is meant, and provide 95% CIs and p values in Table 1 supporting this. “As shown in Table 1, there were no meaningful differences in the characteristics of infants excluded due to lack of EDI testing”

20. Results: Line 236-239: Please include p values along with 95% CIs for the result presented here, and please replace “resulted in” to “associated with” to avoid causal impliacations: “Our regression discontinuity model estimated that the higher rates of antenatal corticosteroid administration before 34+0 weeks were associated with a risk ratio for neonatal respiratory morbidity/mortality of 0.81 [95% CI: 0.72 to 0.91], or 5.3 fewer cases per 100 deliveries [95% CI: 2.5 to 8.3]."

21. Results: Lines 242-249: Please provide both the 95% CIs and p values for all results presented, including the relationships between corticosteroid administration and EDI scores in the discontinuity analysis, and for each of the 5 domains individually. Please revise this sentence to avoid causal implications: “Figure 4a shows that antenatal corticosteroid administration practices was not associated with total EDI scores (median difference of -0.5 [95% CI: -2.1 to 1.2] at the discontinuity of 34+0 weeks; Table 2).”

22. Results: 246-249: Please provide the p values for: “The point estimate for developmental vulnerability was compatible with 246 a small increase in risk (3.8 excess cases per 100 deliveries [95% CI: -1.3 to 10.0]; RR 1.16 [95% CI: 0.92 to 1.42]), but was not statistically significant.”

23.Results: Line 246-249: Please revise to avoid causal implications, and provide p values and 95% CIs with results: “There was no significant association between antenatal corticosteroid administration practices and special needs designation, although the estimates were imprecise (Figures 4b-c).

24. Results: Line 254-257: Please include both the p values and 95% CIs for these findings: “Using multiple imputation to account for missing EDI results produced estimates for total EDI score, developmental vulnerability, and special needs designation were nearly identical to our primary analyses (-0.5 [95% CI: -2.2 to 1.2], 1.16 [95% CI: 0.95 to 1.4] and 1.0 [95% CI: 0.7 to 1.4], respectively).”

25. Discussion: Line 264: Please remove the word “compelling” or replace with “statistically significant” if that is your intended meaning. Please replace “affected” with “were associated with” to avoid causal implications.

26. Discussion: Line 287: Please define the abbreviation ASTECS at first use.

27. Discussion: Line 342: Please replace the term “compliance” with “adherence” where it is used to refer to treatment adherence.

28. Discussion: Please include a discussion of the following points: 

-Please comment on whether the numbers excluded due to lack of gestational age could contribute to any bias in the analysis.

-Please discuss as a limitation the fact that the assumption that steroids are given on hospital admission cannot be tested.

-Please include a discussion that the conclusions relate to steroids given to mothers with preterm birth, and do not apply to women giving birth at >34 weeks or at term, and that neurodevelopmental effects may differ between these two groups.

29. Reporting Checklist: Please ensure that the study is reported according to the STROBE guideline, and include the completed STROBE checklist as Supporting Information. Please add the following statement, or similar, to the Methods: "This study is reported as per the Strengthening the Reporting of Observational Studies in Epidemiology (STROBE) guideline (S1 Checklist)."

If STROBE is not the most appropriate guideline, please choose and report according to the most appropriate guidelines here: http://www.equator-network.org/reporting-guidelines/strobe/

30. Prospective Analysis Plan: Did your study have a prospective protocol or analysis plan? Please state this (either way) early in the Methods section.

c) In either case, changes in the analysis-- including those made in response to peer review comments-- should be identified as such in the Methods section of the paper, with rationale

31.Table 1: Please define abbreviations for BMI and SD in the legend.

32. Table 2: Please define abbreviations for EDI and CI in the legend. Please also present the p values for the risk ratio estimates.

33. Figure 2 legend: Please change “limited” to “limit”

34. Figures 3 and 4: Please specify in the legend that it is the shaded area that indicates the 95% CIs.

35. S1 Table 1: Please define SD and BMI in the legend.

36. S2 Table 2: Please define the abbreviations for EDI and CI in the legend.

Comments from the reviewers:

Reviewer #1: This manuscript reports results of a regression discontinuity study to investigate the effect of corticosteroids on childhood neuro development using data from a population based record linkage study in BC, Canada. The study addresses an important clinical question, the study design is robust and the manuscript itself was a pleasure to read. 

More detailed comments below:

Line 185 - below the below - please change.

The manuscript states that missing EDI values were imputed - what was the definition of the eligible population for whom imputation was carried out? And what approach was used - further information is needed including a comparison of observed and imputed data. Authors should explicitly state that the main analyses used observed and not MI data and provide the rationale for this. 

Figure 1 - it is not clear to me what the full circles indicated. The legend says "coinciding with reduced antenatal corticosteroid administration" - do authors mean the period between 33+4 and 34 weeks? 

Authors note no significant difference in the 1976 trial on line 283. Please can they provide actual estimates to provide further information to readers and avoid the use of arbitrary p-value cut-offs. 

Maternal pre-pregnancy BMI - authors have these data and it is reassuring to see that the distribution is similar across subsets of the dataset. What is the distribution of BMI across gestational age? 

How many child deaths were there (assigned to 0 EDI)?

Line 297 - authors state that observational studies are at increased risk of confounding bias which is absolutely true. But I would point out that the issue is a bit ore nuanced. The current study also analyses observational data - so I would argue that it is the analytical approach that differs in likelihood of confounding bias and would urge authors to use more nuanced language to clarify this. 

Table S2 - please add the ref category, i.e. is a positive difference indicate a higher score in kids treated or untreated. The difference in the total score is somewhat odd - worth double checking? Also, adding the Ns included in each column would be helpful. Finally, there are no details on the criteria/how the model selected optimal window is identified. 

Reviewer #2: The impact of antenatal corticosteroids on later neurodevelopmental outcome is both topical and important. The study addresses this question using an interesting design in a very large and fairly recent linked cohort. The linkage performed and the statistical analysis are impressive. The conclusion that administering steroids at ≤34 weeks does not result in worse neurodevelopmental outcome is perhaps not surprising, but reassuring nonetheless. I have some general and some more specific comments for the authors:

The issue of using gestation of admission as a proxy for gestation of steroid administration is a bit more complex than presented. I appreciate that it is not possible to know when steroids were actually given during an admission, and that the gestation at admission may be the best proxy possible for the time of steroid administration. However, it is not clear that the actual gestation of delivery is known. If this is available, then it would be useful to present some measures of the variance in interval between admission gestation and delivery gestation. There are two points connected with this. Firstly, the gestation of delivery may be more impactful on the neurodevelopmental outcome than the gestation of steroid administration, and the overall conclusions may be affected by the noise introduced by the variance introduced in steroid-to-delivery interval. For some women, for example those admitted to monitor bleeds from a low-lying placenta, there may be a very long interval between admission and delivery, whereas others will arrive and deliver very quickly. Women who arrive and deliver very quickly also present a problem to the analysis, in that the steroids may not have reached efficacy if the administration to delivery interval is short (this is known wrt lung maturation). The admission to delivery interval would perhaps be expected to decrease with increasing gestational age as the risk/benefit ratio of intervening changes - i.e. a 36 week baby may be delivered at first suspicion of a problem, whereas a 31 week baby may provoke more concern and a greater tendency to try to hold off delivering, for example until the germinal matrix is formed. I note that at least a quarter of the cohort were induced or delivered by CS (of which a proportion will be elective or semi-elective). If this is the case, then the error associated with the admission-to-delivery interval will change across the analysis. 

The second point is that if the hypothesis is that the gestation when the brain is first exposed to steroids is the determinant of long-term outcome (which is a reasonable hypothesis and hence the use of admission dates), rather than the timing of delivery/admission, then the statement in lines 137-138 that steroid administration prior to delivery admission is not an important issue is difficult to reconcile. Presumably the risk of having had steroids during a prior admission increases with babies admitted at later gestations, as a function of time-at-risk?

Given that only 50-60% of the cohort at gestations ≤34 weeks actually had steroids, this seems an important point to draw out and discuss. Why is this percentage so low? Was the national guidance regarding steroid administration changed during the study period? Could a sensitivity analysis be performed introducing a dummy variable for babies actually given steroids?

Can the authors further describe the assumption that the impact of gestation of delivery on later neurodevelopmental outcome is smooth - is there an assumption of linearity or any specified form?

Introduction, lines 63-69: it is not really clear how this manuscript, although interesting with regards to the neurodevelopmental impact of steroids given prior to 34 weeks, adds to the body of evidence regarding later steroid administration. Is exposing the fetal brain to steroids at any stage of development equivalent? On further reading, I realise that this point is addressed at the very end of the limitations section. However, it is an odd way to commence the manuscript, given that the data don't address this question.

Methods: Although the linked cohort is very large, the number of babies born preterm per day for whom there is linked data actually works out fairly small at earlier gestations. Was any analysis to investigate power performed?

The breadth and detail of the EDI data available are impressive and comprehensive

Lines 124-127: this seems like an important point to provide a little more information on, as this method has been applied for the majority of the cohort, 2000-2008. Given that the discontinuity is based on a single day difference, could the authors add to the paper the number of observations potentially mis-categorised by this, i.e. the number of patients born prior to 2008, within weeks 33/34?

Line 157-158: is this a score of 1 or 0/10 in each domain? Or is this a cohort-specific or population-specific 10th centile?

Line 169: were all early neonatal deaths included here? Or only those linked to respiratory morbidity?

Line 226: it would be better to replace this single figure of 45% points with the average before v. after 

Line 341: I had understood Figure 1 to be a schematic of hypothetical data? How does it show that a discontinuity exists in the actual data? Can the labelling be a bit more specific about what Figure 1 represents - if it really is just an example of what a regression discontinuity analysis looks like, then it might be better as a supplementary figure, and the much more interesting figure S2 included in the main manuscript instead.

Table 1: inducing a quarter of preterm fetuses is interesting. Can the authors comment on what gestation induction might have been considered/attempted? These are presumably mainly 35-36 weekers. Do you have data on spontaneous v. iatrogenic delivery overall? Could this be added to the useful and well-presented figure S3?

Implications: I think that the first statement here is a really useful and important finding of the study. I'm much less convinced by the extrapolation inherent in the second statement, and I would think about re-working the introduction and discussion more towards the reassurance the study provides regarding neurodevelopmental implications prior to 34 weeks. I realise this may seem a little less 'high impact' - but it is important in its own right.

Reviewer #3: See attachment

Michael Dewey

[LINK]

---

## [Decision Letter · Decision Letter 2]

2 Oct 2020

Dear Dr. Hutcheon,

Thank you very much for re-submitting your manuscript "Antenatal corticosteroid administration and early school age child development: a regression discontinuity study in British Columbia, Canada" (PMEDICINE-D-20-02762R2) for review by PLOS Medicine.

I have discussed the paper with my colleagues and the academic editor and it was also seen again by three reviewers. I am pleased to say that provided the remaining editorial and production issues are dealt with we are planning to accept the paper for publication in the journal.

[LINK]

We look forward to receiving the revised manuscript by Oct 09 2020 11:59PM. 

Sincerely,

Caitlin Moyer, Ph.D.

Associate Editor 

PLOS Medicine

plosmedicine.org

Requests from Editors:

1.Response to Editor’s comment 7 regarding the inclusion of p-values: Thank you for your response. PLOS Medicine policy is to require inclusion of both the 95% CIs and p values for all results. Below, we have noted specific locations in the text where p values should be added. If it is methodologically not possible/appropriate to provide meaningful p-values, please include an explanation (in the methods or a supporting information file) why this particular RD analysis (i.e. the weighting referred to) makes it impossible to report p values. Please also note this where relevant in the Tables where p values can not be provided.

2.Response to reviewers: “We opted not to include our sample size calculation in the

manuscript for conciseness, as our primary analyses had good statistical precision

(demonstrating that under-power/small sample size was not a concern), but are willing

to include this paragraph if requested.”

-Please do add the paragraph describing the sample size calculation and power analysis.

3.Abstract: Methods and Findings: Here and throughout manuscript, it seems like weeks’ (as in weeks of gestation) does not need an apostrophe at the end of the word (unless there is a technical reason for this).

4.Abstract: Please provide the p values associated with: “We found no evidence that antenatal corticosteroid administration practices were linked with altered child development at early school age: median EDI score difference of -0.5 [95% CI: -2.2 to 1.7], RD per 100 births for special needs designation -0.5 [-4.2 to 3.1] and for developmental vulnerability of 3.9 [-2.2, 10.0].”

5.Abstract: Methods and Findings: For the limitations (final sentence) please revise to “A limitation of our study is that the regression discontinuity design estimates the effect of antenatal corticosteroid administration at the gestational age of the discontinuity, 34+0 weeks, so our results may become less generalisable as gestational age moves further away from this point.” or similar to highlight the sentence.

6.Abstract: Conclusions: Please revise to clarify the meaning behind “little evidence” in the first sentence and remove causal language: “Our study did not find that antenatal corticosteroid administration practices were associated with child development at early school age.” or similar

7.Author summary: Why was this study done?: Please combine the third and fourth bullet points: “--Because the breathing complications experienced by preterm births at 34-36 weeks are usually less serious in nature and less common, antenatal care providers are increasingly wanting to know if there are any long-term side-effects of antenatal corticosteroids before administering the medication; however, there is limited information on the longer-term safety of antenatal corticosteroids for overall child developmental health.”

8.Introduction (and throughout the manuscript): While your study design attempts to replicate random allocation, it is not a true randomized trial and they should avoid causal language throughout. Please avoid the use of causal language at lines 127-129 “We estimated the causal effect of antenatal corticosteroid administration by comparing the health outcomes of infants lying just above to those just below the threshold [14].”

9.Methods: For noting statistical significance of results, please specify the significance level used (eg, P<0.05, two-sided) in addition to the statistical test used to derive a p value. Alternatively please explain the rationale for not including p values for certain analyses.

10.For table S1 and Results comparing excluded vs. non-excluded pregnancies: “but there were no clinically important differences in pre-pregnancy body mass index, maternal comorbidities, birthweight, preterm birth, or neonatal outcomes” please specify whether there were statistically significant differences in any of these.

11.Results: bottom of page 11: “As shown in Table 1, there were no clinically important differences in the characteristics of infants excluded due to lack of EDI testing.” Please indicate whether there were any statistically significant differences here.

12.Results: page 12: Neonatal respiratory morbidity and mortality: Please provide p values for the following results: Our regression discontinuity model estimated that the higher rates of antenatal corticosteroid administration before 34+0 weeks resulted in a risk ratio for neonatal respiratory morbidity/mortality of 0.81 [95% CI: 0.71 to 0.92], or 5.4 fewer cases per 100 deliveries [95% CI: 2.3 to 8.8].” In this sentence, please change “resulted in” to “ were associated with” to avoid causal language.

13.Results: page 12: Child development testing: Please update the presentation to include p values to accompany 95% CIs for all reported results.

14.Results: page 13: Sensitivity analyses: Please update the presentation to include p values to accompany the 95% CIs for the reported results using multiple imputation for missing EDI/developmental vulnerability/special needs designation measures.

15. Discussion: Line 321 (and throughout): Please revise to avoid the use of causal language “In this large, population-based cohort, we found no evidence that antenatal corticosteroids administration practices were associated with child development scores at early school age.”

16.Discussion: Line 431: this paragraph would be more appropriately named “conclusions” and a paragraph describing implications and next steps for research, clinical practice, and/or public policy would fall between the paragraphs describing strengths and limitations of your study and the conclusions.

17.Use of first person language (throughout): Please reduce the use of first person language somewhat in the body of the manuscript (some use of first person is fine, particularly in the author summary). Some examples where first person could be reduced are:

-Methods, page 10: Each of these paragraphs starts with “We”

- Discussion Line 432 (and elsewhere): Instead of “Our results provide reassurance that current antenatal corticosteroid practices are not linked…” Please revise to: "These results suggest that current antenatal corticosteroid practices are not linked..."

18.References: Please use the "Vancouver" style for reference formatting, and see our website for other reference guidelines https://journals.plos.org/plosmedicine/s/submission-guidelines#loc-references

Specifically, please check the formatting as it looks like there should be a period between the journal name and the year of publication. Journal name abbreviations should be those found in the National Center for Biotechnology Information (NCBI) databases. (New England Journal of Medicine should be N Engl J Med, for example)

19.Table 1. For birthweight, please indicate the units (grams, for example)

20.Table 2: For each outcome, please provide p values in addition ot the 95% CIs for differences in scores between gestation under vs. equal or over 34 weeks.

21.STROBE checklist: Thank you for including the checklist. The notes in red indicating locations in the text can be difficult to read- can you please add an additional right hand column on the table to designate the locations for the items on the checklist?

Comments from Reviewers:

Reviewer #1: Authors have addressed all comments comprehensively. I have no further comments. 

Reviewer #2: Thank you for this careful revision. The additional analyses included and the clarifications within the manuscript have greatly improved the paper. 

I don't have any further substantive suggestions for the authors.

Reviewer #3: The authors have addressed all my points.

Michael Dewey

[LINK]

---

## [Editor Report · Decision Letter 3]

23 Oct 2020

Dear Dr Hutcheon, 

On behalf of my colleagues and the academic editor, Dr. Sarah J Stock, I am delighted to inform you that your manuscript entitled "Antenatal corticosteroid administration and early school age child development: a regression discontinuity study in British Columbia, Canada" (PMEDICINE-D-20-02762R3) has been accepted for publication in PLOS Medicine. 

PRODUCTION PROCESS

Before publication you will see the copyedited word document (within 5 business days) and a PDF proof shortly after that. The copyeditor will be in touch shortly before sending you the copyedited Word document. We will make some revisions at copyediting stage to conform to our general style, and for clarification. When you receive this version you should check and revise it very carefully, including figures, tables, references, and supporting information, because corrections at the next stage (proofs) will be strictly limited to (1) errors in author names or affiliations, (2) errors of scientific fact that would cause misunderstandings to readers, and (3) printer's (introduced) errors. Please return the copyedited file within 2 business days in order to ensure timely delivery of the PDF proof. 

If you are likely to be away when either this document or the proof is sent, please ensure we have contact information of a second person, as we will need you to respond quickly at each point. Given the disruptions resulting from the ongoing COVID-19 pandemic, there may be delays in the production process. We apologise in advance for any inconvenience caused and will do our best to minimize impact as far as possible.

PRESS

PROFILE INFORMATION

Thank you again for submitting the manuscript to PLOS Medicine. We look forward to publishing it. 

Best wishes, 

Caitlin Moyer, Ph.D.

Associate Editor 

PLOS Medicine

plosmedicine.org